# Planner Metrics Should Satisfy Independence of Irrelevant Alternatives
## (Position Paper)

**Jendrik Seipp**
University of Basel
Basel, Switzerland
jendrik.seipp@unibas.ch

## Abstract

We argue that planner evaluation metrics should satisfy the independence of irrelevant alternatives criterion, i.e., the decision whether planner A is ranked higher or lower than planner B should be independent of planner C. We show that three metrics used in classical planning competitions do not necessarily satisfy this criterion and highlight alternative metrics that do so.

## Introduction

Arrow's impossibility theorem is an important result from social choice theory (Arrow 1950). One of the fairness criteria it suggests is *independence of irrelevant alternatives* (IIA). In the setting of planning competitions, IIA translates to the requirement that the decision whether planner A is ranked higher or lower than planner B must depend only on the performance of planners A and B and not on another planner C. We believe that IIA is a critical requirement for planner evaluation metrics.

In the following, we show three planner evaluation metrics that do not satisfy IIA and give alternative metrics that do satisfy it.

## IPC Satisficing Track

In the satisficing track of the International Planning Competition (IPC) planners are given 30 minutes to find plans. The time for finding plans is ignored, but cheaper plans are preferred. More precisely, the track uses the following metric (which we call *sat*) to evaluate a planner $P$ on task $\pi$: $P$ gets a score of 0 if it fails to solve $\pi$ within the resource limits and a score of $Cost^*/Cost$ if it solves $\pi$, where $Cost$ is the cost of the cheapest plan that $P$ finds for $\pi$ and $Cost^*$ is the cost of a *reference* plan, i.e., a cheapest known plan for $\pi$. The total score for a planner is the sum of its scores over all tasks.

It is easy to see that *sat* satisfies IIA if $Cost^*$ is always the cost of an optimal plan for $\pi$. However, if we take solutions for $\pi$ found by the competing planners into account when computing $Cost^*(\pi)$, *sat* does not satisfy IIA anymore.

We show this claim with the small example in Table 1. The leftmost table shows the cost of the reference plan $R$ and the cost of the plans that planners $A$, $B$ and $C$ find for

| Cost | R | A | B | C | | sat | A | B | | sat | A | B | C |
|------|---|---|---|---|---|------|-----|-----|---|------|-----|-----|-----|
| $\pi_1$ | 2 | 5 | 4 | 5 | | $\pi_1$ | 2/5 | 2/4 | | $\pi_1$ | 2/5 | 2/4 | 2/5 |
| $\pi_2$ | 6 | 4 | 5 | 1 | | $\pi_2$ | 4/4 | 4/5 | | $\pi_2$ | 1/4 | 1/5 | 1/1 |
| | | | | | | $\sum$ | 1.4 | 1.3 | | $\sum$ | 0.65 | 0.7 | 1.4 |

Table 1: Example showing that the evaluation metric of the IPC sequential satisficing track does not satisfy independence of irrelevant alternatives if the reference plans ($R$) are suboptimal.

two tasks $\pi_1$ and $\pi_2$. If only $A$ and $B$ participate in the competition, $A$ achieves a higher *sat* score than $B$ (middle table). However, if $C$ enters the competition as well, $B$ is ranked higher than $A$ (rightmost table).

To mitigate this problem, it is important to use domain-specific solvers to find reference plans of high quality.

## IPC Agile Track

IPC 2014 introduced the *agile* track (Vallati, Chrpa, and Mc-Cluskey 2014). It ignores solution costs and evaluates planners solely by how fast they are able to find a solution. The first agile competition used the following evaluation metric, which we call $agl_{2014}$: for each task $\pi$ in the benchmark set that a planner $P$ solves in under five minutes, $P$ gets a score of $1/(1 + \log_{10}(T/T^*))$, where $T$ is the time $P$ needs for solving $\pi$ and $T^*$ is the minimum runtime of any participating planner. As in the satisficing track, the total score for a planner is the sum of its scores over all tasks.

Clearly, $agl_{2014}$ does not satisfy IIA, which is the reason the agile track used a different evaluation metric in 2018. The 2018 metric, which we call $agl_{2018}$, evaluates each planner on its own. If $T$ is the time in seconds a planner $P$ needs to solve task $\pi$,[1] $P$ gets the score $agl_{2018}(P, \pi)$, which is defined as

$$agl_{2018}(P, \pi) = \begin{cases} 0 & \text{if } T > 300 \\ 1 & \text{if } T < 1 \\ 1 - \frac{\log(T)}{\log(300)} & \text{if } 1 \leq T \leq 300 \end{cases}$$

---

[1]We define $T$ to be $\infty$ if the planner exceeds the memory limit.

It is easy to see that $agl_{2018}$ indeed satisfies IIA and is therefore preferable to $agl_{2014}$ in our opinion.

## Sparkle Planning Challenge

In 2019, the Sparkle Planning Challenge will be held for the first time. Its "primary goal [...] is to analyse the contribution of each planner to the real state of the art".[2] Consequently, the challenge measures the marginal contribution of each participating planner to a portfolio selector, i.e., an algorithm that chooses a single planner from the set of all participating planners online for a given task. In essence, the Sparkle evaluation metric, which we call *sparkle*, evaluates a planner $P$ by the penalized average runtime the portfolio selector achieves when it can select from all competing planners except $P$.

The following example shows that *sparkle* does not satify IIA. Assume planners $A$ and $B$ enter the Sparkle Planning Challenge and the benchmark set consists of 100 tasks. Planner $A$ solves a single task $\pi$ within the time and memory limits while planner $B$ solves the other 99 tasks but fails to solve $\pi$. Clearly, $B$ wins the competition. Now imagine that planner $C$ also participates in the challenge. Planner $C$ solves the same tasks as $B$ and has almost the same runtimes. Under *sparkle*, $B$ and $C$ contribute almost nothing to the portfolio (since without $B$ the selector can still choose $C$ and vice versa) and planner $A$ wins the challenge.

We believe there is no reason for penalizing planners B and C for performing similarly. In fact, we consider *sparkle* to be quite problematic, since a scenario similar to the one in the example above is quite likely to come up in competitions and we see mainly two reasons for why it might.

First, the evaluation metric is easily gameable. If a team wants to win the challenge, it just needs to guess which other planners might be participating and submit similar planners in addition to the "real" planner. By the competition rules, a team of three can submit one real planner and 6 "fake" planners. Having the leader board available online before the submission deadline and the IPC 2018 planners readily available makes the metric easy to exploit.

Second, and we think this reason is even more important than the first one, the evaluation metric penalizes collaboration. It favors developing closed-source planning systems over developing planning systems openly and allowing others to build on the system. This is the case, since all planners that share the same base planning system are likely to perform well on similar tasks and therefore receive low marginal contribution scores. Openly developed planning systems have a disadvantage in the IPC as well. However, under IPC metrics a planner that builds on an open planning system might score slightly higher than the system, but under the Sparkle Challenge metric both would get a very low score.

An evaluation metric that satisfies IIA does not suffer from these problems. In order to let *sparkle* satisfy IIA, we only need to change it slightly. Instead of evaluating a planner with respect to all participating planners, we can evaluate it with respect to a fixed set of baseline planners. We believe that this set of baseline planners should be accumulated over time. A reasonable first set of baseline planners could be the set of all planners from the last IPC agile track. Subsequent challenges should add planners from later agile tracks and/or Sparkle planning challenges.

## Acknowledgments

We thank the anonymous reviewers for their helpful comments. We have received funding for this work from the European Research Council (ERC) under the European Union's Horizon 2020 research and innovation programme (grant agreement no. 817639).

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
