# OpenReview forum: "Planner Metrics Should Satisfy Independence of Irrelevant Alternatives"
_icaps-conference.org/ICAPS/2019/Workshop/WIPC_

### Official Review · AnonReviewer2 · 2019-04-25
**Review of this position paper**

**Rating:** 8
**Confidence:** 5

**Review:**

This position paper argues that IPC metrics should satisfy independence of irrelevant alternatives.  As well as discussing this in the context of traditional IPCs, the paper also notes that in the sparkle challenge, two similar planners will receive low scores even if they are important to the best portfolio.

I think this is a good addition to the workshop, and it would be helpful to have community input on how planners (and planners in portfolios) should be reasonably compared.

Other notes:

P1 - bottom left - the term 'reference plan' needs to be defined

P2 - lhs - 'too be quite' -> 'to be quite'
P2 - lhs - 'Openly-developed' -> 'Openly developed'

---

### Official Review · AnonReviewer3 · 2019-04-26
**Well-argued opinion paper**

**Rating:** 8
**Confidence:** 4

**Review:**

This paper argues that the results of the competition should satisfy independence of
irrelevant alternatives. In other words, the relative comparison of two planners should
not be affected by other participants. The authors introduce three different cases where
tracks of the IPC and related competitions did not satisfy this assumption and discusses
alternative scores that would satisfy it.

This is a well-argued opinion paper that is very relevant to the discussion in the
workshop. My 2 cents:

  - As problematic consequences of the criteria chosen in the sparkle challenge, you can
    also add that discourages participation. Since submitting "related" approaches will
    reduce the score of all of them, no research group should submit more than one
    planner. This makes the competition less interesting since planners that could
    potentially show interesting results at least in some domains won't be submitted.

  - I like the proposed way to fix the sparkle challenge, but this has also some
    issues. In particular, the results will be very heavily influenced by the set of
    "baseline" planners, which is supposed to be representative of the "state of the
    art". In particular, it would penalize the authors of baseline planners whereas anyone
    whose planner was not chosen would have a huge advantage. Choosing the planners from
    the latest Sparkle challenge may seem a good idea to avoid unfair biases. However,
    this requires everyone in the community having participated in the previous editions
    of the sparkle challenge, which does not seem to be the case (around 20 planners
    participated in IPC'18, while currently there are 4-5 systems in this year's
    challenge). Including historic planners may help but it would also significantly
    increase the workload of the organizers.

Minor comments:
 reson -> reason
 too be quite -> to